# Increased Density of Endogenous Adenosine A_2A_ Receptors in Atrial Fibrillation: From Cellular and Porcine Models to Human Patients

**DOI:** 10.3390/ijms24043668

**Published:** 2023-02-11

**Authors:** Héctor Godoy-Marín, Verónica Jiménez-Sábado, Carmen Tarifa, Antonino Ginel, Joana Larupa Dos Santos, Bo Hjorth Bentzen, Leif Hove-Madsen, Francisco Ciruela

**Affiliations:** 1Pharmacology Unit, Department of Pathology and Experimental Therapeutics, School of Medicine and Health Sciences, Institute of Neurosciences, University of Barcelona, 08907 L’Hospitalet de Llobregat, Spain; 2Neuropharmacology & Pain Group, Neuroscience Program, Bellvitge Institute for Biomedical Research, 08907 L’Hospitalet de Llobregat, Spain; 3Biomedical Research Institute Sant Pau, IIB Sant Pau, 08025 Barcelona, Spain; 4Centro de Investigación Biomédica en Red Enfermedades Cardiovasculares, 28029 Madrid, Spain; 5Biomedical Research Institute of Barcelona, IIBB-CSIC, 08036 Barcelona, Spain; 6Department Cardiac Surgery, Hospital de la Santa Creu i Sant Pau, 08036 Barcelona, Spain; 7Department of Biomedical Sciences, University of Copenhagen, DK-2200 Copenhagen, Denmark

**Keywords:** adenosine A_2A_R receptor, atrial fibrillation, HL-1 cells, tachypaced pig, A_2A_R density, AF patients

## Abstract

Adenosine, an endogenous nucleoside, plays a critical role in maintaining homeostasis during stressful situations, such as energy deprivation or cellular damage. Therefore, extracellular adenosine is generated locally in tissues under conditions such as hypoxia, ischemia, or inflammation. In fact, plasma levels of adenosine in patients with atrial fibrillation (AF) are elevated, which also correlates with an increased density of adenosine A_2A_ receptors (A_2A_Rs) both in the right atrium and in peripheral blood mononuclear cells (PBMCs). The complexity of adenosine-mediated effects in health and disease requires simple and reproducible experimental models of AF. Here, we generate two AF models, namely the cardiomyocyte cell line HL-1 submitted to Anemonia toxin II (ATX-II) and a large animal model of AF, the right atrium tachypaced pig (A-TP). We evaluated the density of endogenous A_2A_R in those AF models. Treatment of HL-1 cells with ATX-II reduced cell viability, while the density of A_2A_R increased significantly, as previously observed in cardiomyocytes with AF. Next, we generated the animal model of AF based on tachypacing pigs. In particular, the density of the key calcium regulatory protein calsequestrin-2 was reduced in A-TP animals, which is consistent with the atrial remodelling shown in humans suffering from AF. Likewise, the density of A_2A_R in the atrium of the AF pig model increased significantly, as also shown in the biopsies of the right atrium of subjects with AF. Overall, our findings revealed that these two experimental models of AF mimicked the alterations in A_2A_R density observed in patients with AF, making them attractive models for studying the adenosinergic system in AF.

## 1. Introduction

Atrial fibrillation (AF) is the most common cardiac arrhythmia, currently thought to affect approximately 2% of the world’s population. Its prevalence increases with age, affecting approximately 0.14% of people under 49 years of age and increasing to 10–17% in people 80 years or older [1]. Among the molecular mechanisms believed to contribute to the development of AF, stress-mediated activation of G protein-coupled receptors has been associated with modulation of ion currents leading to the development of electrical instability. This includes: (i) elevation of sympathetic tone by activation of beta-adrenergic receptors; (ii) hyperphosphorylation of calcium regulatory proteins, causing calcium release-induced afterdepolarizations and triggered activity [1]; and (iii) parasympathetic activation of G_αi_ protein-coupled receptors triggering potassium currents that shorten the refractory period, thus favouring electrical re-entry and arrhythmia [1,2,3]. Interestingly, similar effects can be induced by activation of adenosine receptors (ARs) [4,5] and we have recently documented that AF is associated with elevated adenosine plasma levels (APLs) [6], which, combined with increased expression and activation of adenosine A_2A_ receptors (A_2A_Rs) in AF [6,7], may induce hyperphosphorylation of the cardiac ryanodine receptor (RyR2), spontaneous calcium release, afterdepolarizations, and irregular beating after elevation of the stimulation frequency observed in the atrial myocytes of patients with AF [7,8]. On the other hand, elevated APLs in AF might also activate adenosine A_1_ receptors (A_1_Rs) and trigger the atrial acetylcholine-activated inward rectifier potassium current (I_KACh_) reported in AF [9,10,11] that reduces the refractory period and facilitates electrical re-entry, which in turn may contribute to the development and maintenance of AF. Therefore, the functional impact of elevated APLs in AF probably depends on the binding of adenosine to atrial A_1_R and A_2A_R.

Experimental models of AF are commonly used to study the underlying mechanisms and evaluate potential treatments for the disease. There are several types of experimental models used in AF research, including: (i) animal models based on the controlled induction of AF in various species, such as pigs, dogs, and mice; (ii) in vitro models mostly using isolated cardiomyocytes or tissue slices from several sources, including humans; and (iii) computational models simulating the electrical and structural properties of the heart. Each AF model has its own advantages and limitations, and the choice of model depends on the scientific question under study. Some of the limitations of AF models comprise the lack of complete representation, which limits their predictive value, the fibrillation inducing method, which may not accurately reflect the underlying mechanisms of human AF. In addition, there are species differences (i.e., anatomy and physiology), which limit the generalisability of results from animal models to humans. Finally, the limited availability of human tissue due to ethical considerations and the difficulty of obtaining fresh samples may also be a drawback.. Despite these limitations, AF experimental models remain an important tool for understanding the underlying mechanisms of the disease and evaluating potential treatments.

Here, several experimental models of AF, namely a stable cardiomyocyte cell line (that is, HL-1 cells), a pig animal model of AF, and human biopsies from patients with AF, were implemented to assess the density of A_2A_R in AF.

## 2. Results

### 2.1. Endogenous A_2A_R Density in the HL-1 Cell Line

HL-1 cells are a commonly used cell line in cardiac research due to their differentiated cardiac phenotype and spontaneous electrical and contraction activity. Endogenous AR expression has been characterised in these cells along with many other cardiac characteristics [12]. Therefore, our objective was to determine A_2A_R density in HL-1 cells under basal and in vitro-induced arrythmia-like conditions. To create an arrhythmia-like condition in HL-1 cells, we applied Anemonia toxin II (ATX-II), a toxic peptide derived from the coelenterate *Anemonia viridis*. ATX-II modulates voltage-gated sodium channel activity, extending the duration of the excitable cell action potential and simulating an arrhythmia-like state [13]. The impact of the ATX-II-mediated arrhythmia-like condition of HL-1 cells was measured by three previously used assays [14]. To this end, we implemented 3-(4,5-dimethylthiazol-2-yl)-2,5-diphenyltetrazolium bromide (MTT), propidium iodide (PI) and dichlorodihydrofluorescein (DCF) assays that determine cell viability, cell damage, and ROS production, respectively, in HL-1 cells incubated in the absence and presence of ATX-II (30 nM) for 24 h. Treatment with ATX-II significantly reduced cell viability and increased cell damage and ROS production (Figure 1), confirming that ATX-II incubation compromises membrane potential and favours a pro-arrhythmic state in HL-1 cell culture.

After inducing the arrythmia-like state in HL-1 cells, our objective was to determine the impact of ATX-II treatment on endogenous A_2A_R expression. To this end, we performed immunoblot experiments to assess A_2A_R density in membrane extracts of HL-1 cells incubated in the absence and presence of ATX-II (30 nM) for 24 h (Figure 2). Interestingly, when HL-1 cells were treated with ATX-II a significant increase (53 ± 19%; *p* = 0.027) in the density of endogenous A_2A_R was observed (Figure 2). Therefore, these results indicated that the ATX-II-induced arrhythmia-like condition in HL-1 cells increased A_2A_R expression. In general, our results demonstrated that in the ATX-II-induced arrhythmia-like cell model, A_2A_R density increases significantly, suggesting a potential deregulation of adenosine signalling through a concomitant imbalance of ARs.

### 2.2. Expression of A_2A_R in the Atrium of a Porcine Model of AF

Pigs have been used for a long time in cardiovascular research due to their advantages over other animal models such as dogs, sheep, mice, and rats [15]. Consequently, we used cardiac tissue from a porcine model with atrial tachypacing (A-TP), as described in [15]. Therefore, atrial membrane extracts from the control group of sham-operated pigs (SHAM) and A-TP pigs were prepared and analysed by immunoblot. First, to assess the degree of atrial remodelling within our porcine model of AF, we evaluated the relative density of the calcium handling proteins SERCA2a, NCX-1, and Csq-2 (Figure 3A), since an alteration of these proteins has been suggested to occur in AF [16]. Interestingly, although no significant alteration was found in the densities of atrial SERCA2a (*p* = 0.682) and NCX-1 (*p* = 0.952) in A-TP pigs, a significant reduction in the density of atrial Csq-2 (*p* = 0.01) was demonstrated (Figure 3B). Importantly, we recently reported an alteration in the relative expression of these calcium-handling proteins in atrial membrane extracts from patients with AF [17]. To be precise, while SERCA2a and NCX-1 density was not altered in patients with AF, Csq-2 expression was significantly lower in subjects with AF compared to that in subjects with nondilated sinus rhythm (ndSR), which were used as a non-AF control group [17]. In general, our results in A-TP pigs showing a significant reduction in Csq-2 expression after AF induction are in line with those found in patients with AF, suggesting the existence of a common remodelling of intracellular Ca^2+^ handling (that is, reduction of Csq-2) in AF, which validates our porcine animal model of AF.

Subsequently, our objective was to determine the density of A_2A_R in the atrium of SHAM and A-TP pigs by immunoblotting experiments (Figure 4A). Interestingly, our immunoblot results revealed that the density of A_2A_R increased significantly (73 ± 22%; *p* = 0.011) in A-TP pigs compared to SHAM animals (Figure 4B). In general, these results indicating that A_2A_R expression in the atrium of A-TP animals is enhanced are in line with those obtained in HL-1 cells, validating the usefulness of these models to study the imbalance of the adenosinergic system in AF.

### 2.3. Expression of A_2A_R in the Atrium of Patients with AF

After we determined the density of A_2A_R in the arrhythmia-like cellular model and the porcine animal model of AF, our aim was to contrast these results with cardiac tissue from patients with AF (Table 1). We have previously demonstrated that A_2A_R expression increases in patients with AF [6,7]. However, we again evaluated the density of A_2A_R in the atrium of patients with AF (Table 1). To this end, we performed immunoblot experiments (Figure 5A) to determine A_2A_R density in atrial membranes from ndSR subjects and patients with AF. Again, we confirm a significant increase (58 ± 13%; *p* = 0.0004) in the density of atrial A_2A_R in patients with AF compared to that in patients with ndSR (Figure 5B).

Overall, we confirmed that the density of A_2A_R in three different AF models increased, confirming the potential usefulness of these models to study AF.

## 3. Discussion

AF has been associated with dysregulation of the adenosinergic system. Specifically, increased APLs and reduced adenosine deaminase (ADA) activity have been demonstrated in subjects with AF [6]. Furthermore, increased expression of A_2A_R has been shown in both the right atrium and peripheral blood mononuclear cells (PBMCs) in the same patients [6,7]. Functionally, increased expression and activation of A_2A_R in atrial myocytes from AF patients has been shown to stimulate spontaneous calcium release, electrical activity [7], and irregular beating [8]. The increased spontaneous calcium release in human atrial myocytes linked to A_2A_R is related to an increased co-distribution of the receptor and RyR2 [4], which can be reduced using A_2A_R antagonists or adenosine deaminase [4,17]. Therefore, we questioned whether this dysregulation of the adenosinergic system constitutes a pathological hallmark of AF and whether this could also be mirrored in experimental models of AF. Consequently, we monitored the density of A_2A_R in two experimental AF models (that is, HL-1 cells treated with ATF and A-TP pigs) while comparing them to right atrial biopsies from human AF patients. Our results demonstrated that in all three models of arrhythmias, the density of A_2A_R was significantly increased, indicating a deregulation of purinergic signalling in AF.

In vitro assays and animal models of disease constitute essential tools for biomedical investigations of pathological conditions. In cellular models of AF, electrical stimulation of primary cultured cardiomyocytes is generally used. However, due to the limited access to human biopsies suitable for the successful isolation of viable myocytes, its use has been diminishing over the years. Interestingly, these inherent problems can be overcome by using an immortalised cardiomyocyte cell line, such as the HL-1 cell line, which has been widely used as a model to study cardiovascular diseases, including AF [18]. Here, we implemented and expanded the HL-1 cell-based model to study AF. Specifically, we altered sodium influx in HL-1 cells to induce an arrhythmic state similar to that observed in AF cardiomyocytes. For this, we used ATX-II, a cardiotoxic peptide [13,19] that modifies the voltage-gated Na^+^ channel kinetics of HL-1 cells, leading to an excitotoxic pro-arrhythmic state that can be easily determined by assessing metabolic activity, membrane integrity and ROS formation in HL-1 cells. Accordingly, the incubation of HL-1 cells with 30 nM ATX-II for 24 h significantly affected all three indicators of an excitotoxic condition, thus validating this in cellulo model of AF. In general, since the ATX model shows changes in A_2A_R expression that are comparable to those observed in AF, it could become a promising tool to evaluate candidate compounds targeting A_2A_R overactivity.

Several animal models have been developed for cardiac research [20]. Classically, the use of small rodents (i.e., mice and rats) is the most common approach. However, small rodents show substantial differences in scale and cardiac electrophysiology compared to humans. In general, the size of the cardiac chambers makes it difficult to naturally maintain the AF phenotype, and the heart rate is up to ten times higher in small rodents, making extrapolations from models to humans complicated. Concerning the use of larger animals, the most common models are goats, dogs and sheep, but the pig animal model of AF has gained importance because it offers several advantages: (i) pigs have anatomy and physiology similar to that of humans, which makes their hearts suitable for studying human heart conditions; (ii) porcine AF models have a high degree of reproducibility, making them ideal for use in preclinical trials and other research studies; (iii) development of AF is achievable through a variety of methods, such as rapid atrial pacing, transoesophageal stimulation, or surgical modification; (iv) the size of the porcine heart makes it possible to perform various interventional procedures, such as catheter ablation, in a manner similar to that done in humans; and (v) pigs have a longer lifespan compared to other laboratory animals, allowing longer-term studies of AF. These factors make the porcine model a valuable tool for researchers studying AF and eventually for testing candidate compounds targeting the adenosinergic system. Interestingly, our A-TP pigs showed alterations in the density of atrial calcium-handling proteins like those described in atrial samples from patients with AF [17,21]. Specifically, Csq-2 levels were significantly lower than in patients with AF. Since this protein is the key SR calcium buffer, its reduction can lead to calcium overload in the SR lumen and prompt spontaneous calcium release in the AF.

Overall, the different experimental models of AF may have different arrhythmic phenotypes, as the underlying mechanisms and conditions that trigger the onset of AF can vary between models. Indeed, this can affect the features and characteristics of the arrhythmia-like phenomena observed in each model, such as the frequency, duration, and stability of the fibrillation waves. Therefore, it is important to carefully choose the appropriate experimental model to study the specific aspects of AF that are of interest. Our results revealed that both HL-1 arrhythmic cells and A-TP pigs showed an increased density of A_2A_R, exactly as occurs in patients with AF [6,7]. However, it remains to be elucidated whether this increase in A_2A_R expression may be the underlying cause or secondary to the development of AF.

## 4. Materials and Methods

### 4.1. Reagents

The following reagents were used: ATX-II (Alomone Labs, Jerusalem, Israel). The antibodies used were mouse anti-A_2A_R (sc-32261, Santa Cruz Biotechnology Inc.), rabbit anti-α-actinin (sc-17829; Santa Cruz Biotechnology Inc.), mouse anti-sarco/endoplasmic reticulum calcium-ATPase 2 (SERCA2a) (D51B11, Cell Signaling Technlogy, Danvers, MA, USA), rabbit anti-Na^+^/Ca^2+^ exchanger 1 (NCX-1) (ab135735, Abcam, Cambridge, United Kingdom), rabbit anti-calsequestrin-2 (Csq-2) (ab3516, Abcam), horseradish peroxidase (HRP)-conjugated rabbit anti-mouse IgG (Pierce Biotechnology, Rockford, IL, USA) and HRP-conjugated goat anti-rabbit IgG (Pierce Biotechnology, Rockford, IL, USA).

### 4.2. Cell Culture and Viability Assays

The mouse cardiomyocyte cell line HL-1 [12] was grown in claycomb medium (Sigma-Adrich) supplemented with 10% FBSi (*v*/*v*), 100 U/mL streptomycin, 100 mg/mL penicillin, 0.1 mM norepinephrine and 2 mM L-glutamine at 37 °C and 5% CO_2_. The absence of mycoplasma was checked regularly to avoid contamination by mycoplasma.

Three different viability assays were performed to test the effect of ATX-II on HL-1 cells: (i) 3-(4,5-dimethylthiazol-2-yl)-2,5-diphenyltetrazolium bromide (MTT) assay to assess cell metabolic activity; (ii) propidium iodide (PI) assay to evaluate membrane integrity and monitor cell damage; and (iii) dichlorodihydrofluorescein (DCF) assay to monitor ROS production. Consequently, HL-1 cells were cultured in 6-well plates and incubated in the presence of 50 nM ATX-II for 24 h. The cells were then collected and incubated with MTT (MTT Assay for Cell Viability and Proliferation, Sigma), PI (100 µg/mL) and DCF (1 µM), following the manufacturer’s indications. Then, the absorbance of MTT at 570 nm and the fluorescence of the PI (610 nm) and DCF (530 nm) assays were recorded on a CLARIOstar microplate reader (BMG Labtech, Ortenberg, Germany).

### 4.3. Porcine Model of AF

Atrial biopsies from a porcine model of AF were used. For a detailed description of the right atrium tachypaced pig (A-TP) model, see [15]. A total of 6 Danish landrace gs (gilts) had cardiac pacing devices implanted at an age of 11 weeks (30–35 kg), which constituted the AF-induced A-TP group, and 4 pigs were sham-manipulated and used as the control group. The A-TP pigs were tachypaced for 35 days. Sustained AF occurred after 17 ± 2 days.

### 4.4. Human Samples and Subject Demographics

Human right atrial heart tissue was collected from 20 patients undergoing cardiac surgery at the Hospital de la Santa Creu i Sant Pau in Barcelona. Of these 20 patients, 9 of them had atrial fibrillation (AF) and the remaining 11 individuals constituted a group of non-dilated sinus rhythm (ndSR) patients. Tissue samples were obtained from the right atrial appendage just before atrial cannulation in surgeries that required extracorporeal circulation. The samples were transported in cold oxygenated Tyrode solution to the laboratory, where they were stored immediately at −80 °C. Each patient gave his/her written consent for the extraction of a sample from the right atrial appendix that would otherwise have been discarded during the surgical intervention. Baseline demographic characteristics and echocardiographic data are summarised in Table 1. Patients who received mitral valve replacement or repair were not included in this study to avoid the confounding effects of mitral valve disease, since this has previously been shown to alter calcium homeostasis in patients without AF [22]. The study protocol was approved by the Ethics Committee of the Hospital de la Santa Creu i Sant Pau, Barcelona, Spain, and the investigation conforms to the principles outlined in the Declaration of Helsinki.

### 4.5. Preparation of Membrane Extracts

HL-1 cells were resuspended in ice-cold 10 mM Tris HCl, pH 7.4 buffer containing a protease inhibitor cocktail (Roche Molecular Systems, Belmont, CA, USA). Cells were homogenized using a Polytron (VDI 12, VWR, Barcelona, Spain) for three periods of 10 s each. The homogenate was centrifuged at 12,000× *g* at 4 °C for 30 min. The membranes were dispersed in 50 mM Tris HCl (pH 7.4). The protein concentration was determined using the bicinchoninic acid (BCA) protein assay kit (Thermo Fisher Scientific, Inc., Rockford, IL, USA).

The right atrial pig and human samples were pulverised in liquid nitrogen congelation and sonicated (Brandson Sonifier 250, ICN Hubber S.A.) in 500 μL of ice-cold 10 mM Tris HCl, pH 7.4 buffer containing a protease inhibitor cocktail [6]. The sonicated tissue was further homogenised using a Polytron (VDI 12) for three periods of 10 s each and processed as described above. The protein concentration was determined using the BCA protein assay kit.

### 4.6. Gel Electrophoresis and Immunoblotting

Sodium dodecyl sulphate polyacrylamide gel electrophoresis (SDS/PAGE) was performed using 10% polyacrylamide gels. Proteins were transferred to Hybond^®^-LFP polyvinylidene difluoride (PVDF) membranes (GE Healthcare, Chicago, IL, USA) using the Trans-Blot^®^ TurboTM transfer system (Bio-Rad, Hercules, CA, USA) at 200 mA/membrane for 30 min. The PVDF membranes were blocked with 5% dry, non-fat milk (wt/vol) in phosphate buffered saline (PBS; 8.07 mM Na_2_HPO_4_, 1.47 mM KH_2_PO_4_, 137 mM NaCl, 0.27 mM KCl, pH 7.2) containing 0.05% Tween-20 (PBS-T) for 1 h at 20 °C before immunoblotting with the indicated antibody in blocking solution overnight at 4 °C. The PVDF membranes were washed with PBS-T three times (5 min each) before incubation with HRP-conjugated rabbit anti-mouse IgG (1/10,000) or HRP-conjugated goat anti-rabbit IgG (1/30,000) in blocking solution at 20 °C for 2 h. After washing the PVDF membranes with PBS-T three times (5 min each), the immunoreactive bands were developed using a chemiluminescent detection kit (Thermo Fisher Scientific) and detected with an Amersham Imager 600 (GE Healthcare Europe, Barcelona, Spain).

### 4.7. Data and Statistical Analysis

Data are represented as mean ± standard error of mean (SEM) with statistical significance set at *p* < 0.05. The number of samples/subjects (*n*) in each experimental condition is indicated in the corresponding figure legend. Outliers were evaluated using the ROUT method [23], and subjects were excluded assuming a Q value of 1% in GraphPad Prism 9 (San Diego, CA, USA). Comparisons between experimental groups were made using a Student’s t test or one-way analysis of variance (ANOVA) followed by Dunnett’s multiple comparison post hoc tests using GraphPad Prism 9, as indicated.

## Figures and Tables

**Figure 1 ijms-24-03668-f001:**
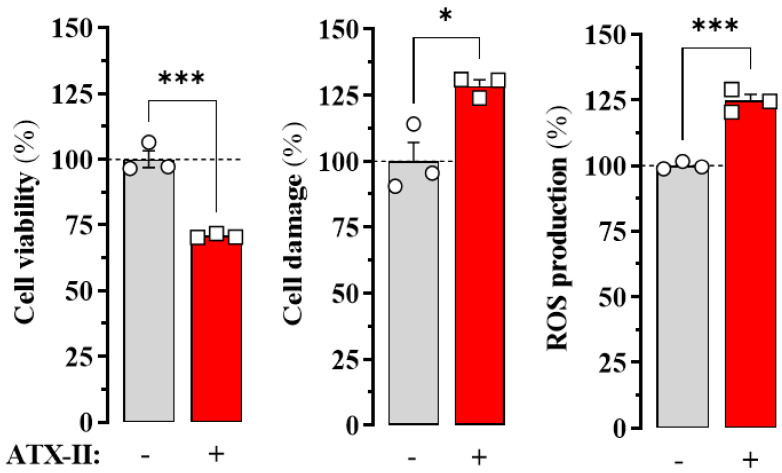
Effect of ATX-II on the viability of HL-1 cells. Cells were incubated in the absence (circles in grey columns,) or presence of ATX-II (30 nM) (squares in red columns) for 24 h before cell viability, cell damage and ROS production were monitored by MTT tetrazolium reduction (left panel), PI (middle panel), and DCF (right panel) assays, respectively (see Methods). The results are shown as percentage of viable cells, cell damage and ROS production in cells treated with saline (ATX-II -) and expressed as mean ± SEM of three independent experiments, each performed in triplicate. * *p* < 0.05, *** *p* < 0.001, Student’s *t*-test.

**Figure 2 ijms-24-03668-f002:**
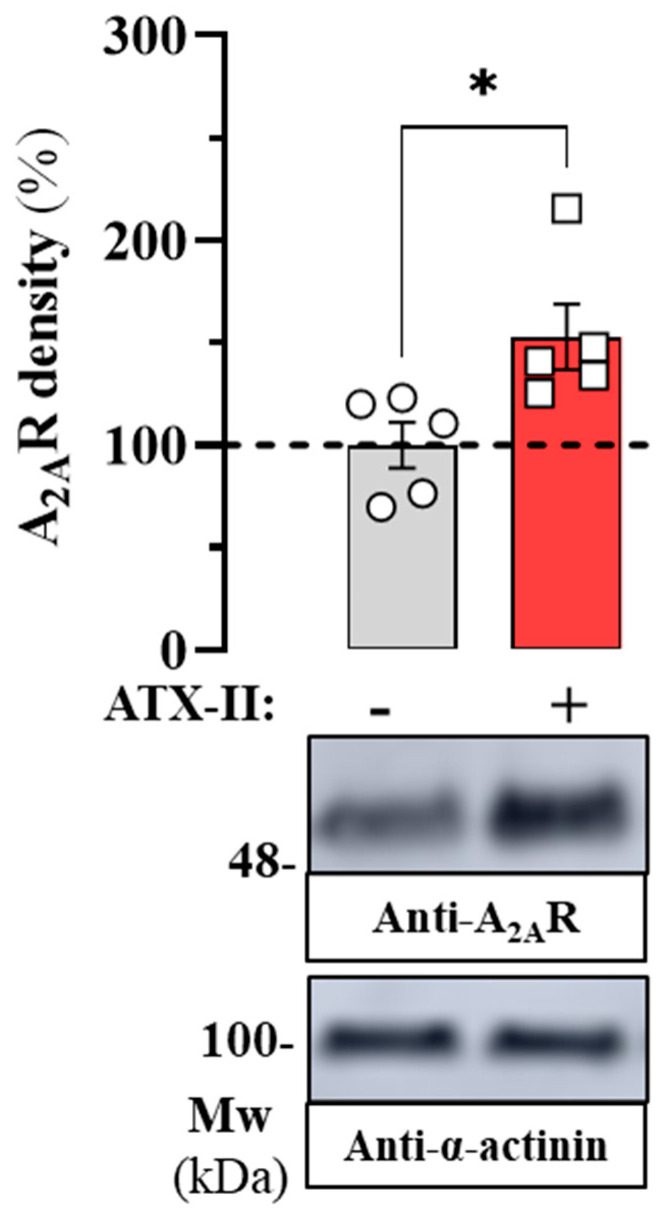
Effect of ATX-II on the density of A_2A_R in HL-1 cells. HL-1 cells were incubated in the absence (circles in grey colum) or presence of ATX-II (30 nM) (squares in redcolumn) for 24 h before membrane extracts were analysed by SDS-PAGE (20 μg of protein/lane) and immunoblotted using mouse anti-A_2A_R and rabbit anti-α-actinin antibodies. A representative immunoblot is shown (lower panel). Immunoblot protein bands corresponding to A_2A_R and α-actinin from vehicle (*n* = 5) and ATX-II (*n* = 5) treated cells were quantified by densitometric scanning. Values were normalized to the respective amount of α-actinin in each lane to correct for protein loading. The results are shown as percentage of the relative density of A_2A_R in cells treated with saline (ATX-II -; dashed lane) and expressed as mean ± SEM (*n* = 5). * *p* < 0.05, Student’s *t*-test.

**Figure 3 ijms-24-03668-f003:**
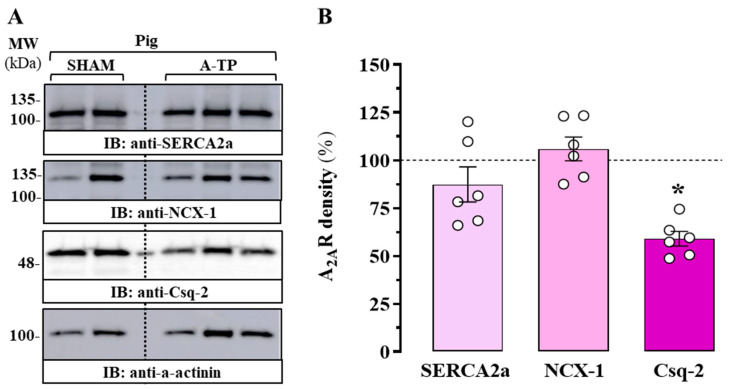
Expression of calcium-handling proteins in the atrium of the porcine model of AF. (**A**) Representative immunoblots showing the density of SERCA2a, NCX-1, and Csq-2 proteins in the right atrium of control (SHAM) and tachypaced (A-TP) pigs. The atrial membrane extract of SHAM and A-TP pigs was analysed by SDS-PAGE (10 μg of protein/lane) and immunoblotted using antibodies against SERCA2a, NCX-1, and Csq-2 (see Methods). (**B**) Relative quantification of SERCA2a, NCX-1, and Csq-2 density. Immunoblot protein bands corresponding to SERCA2, NCX-1, and Csq-2 and α-actinin from SHAM (*n* = 4) and A-TP pigs (*n* = 6) were quantified by densitometric scanning. The values were normalized to the respective amount of α-actinin in each lane to correct for protein loading. The results are shown as percentages of the relative density of each calcium-handling protein in A-TP animals relative to control pigs (SHAM; dashed line) and are expressed as mean ± SEM (*n* = 6). * *p* < 0.05 one-way ANOVA with Dunnett’s post hoc test compared to control pigs (SHAM; dashed line).

**Figure 4 ijms-24-03668-f004:**
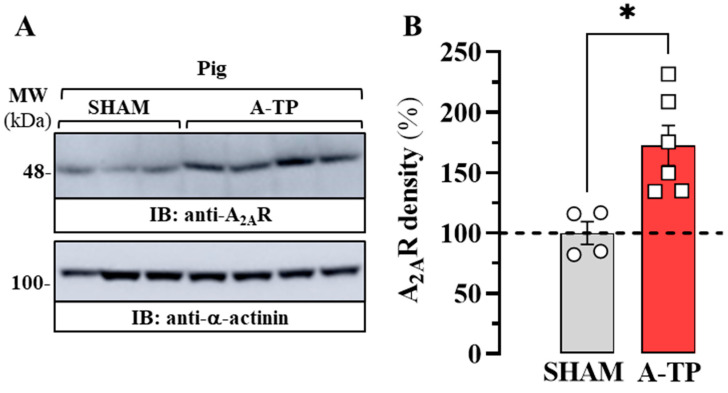
Expression of A_2A_R in the atrium of the porcine model of AF. (**A**) Representative immunoblot showing the density of A_2A_R in the right atrium of control (SHAM, circles) and tachypaced (A-TP, squares) pigs. The atrial membrane extract of SHAM and A-TP pigs was analysed by SDS-PAGE (10 μg of protein/lane) and immunoblotted using mouse anti-A_2A_R and rabbit anti-α-actinin antibodies (see Methods). (**B**) Relative quantification of A_2A_R density. The immunoblot protein bands corresponding to A_2A_R and α-actinin from SHAM (*n* = 4) and A-TP pigs (*n* = 6) were quantified by densitometric scanning. Values were normalized to the respective amount of α-actinin in each lane to correct for protein loading. The results are shown as percentages of the relative density of A_2A_R in SHAM pigs (dashed lane) and expressed as mean ± SEM. * *p* < 0.05, Student’s *t*-test.

**Figure 5 ijms-24-03668-f005:**
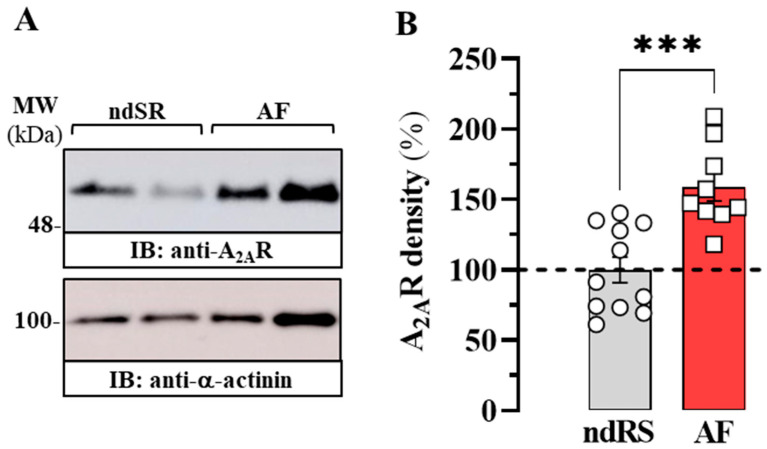
Expression A_2A_R in human heart atria of AF patients. (**A**) Representative immunoblot showing the expression of the A_2A_ receptor in the right atria of non-dilated sinus rhythm (ndSR, circles) and atrial fibrillation (AF, squeres) patients. Membrane extracts from human atrium were analysed by SDS-PAGE (10 μg of protein/lane) and immunoblotted using rabbit anti-A_1_R, mouse anti-A_2A_R and rabbit anti-α-actinin antibodies (see Material and Methods). (**B**) Relative quantification of A_2A_R density. The immunoblot protein bands corresponding to A_2A_R and α-actinin from ndSR (*n* = 11) and AF (*n* = 9) subjects were quantified by densitometric scanning. Values were normalized to the respective amount of α-actinin in each lane to correct for protein loading. The results are shown as percentage of the relative density of A_2A_R in ndRS (dashed lane) and expressed as mean ± SEM. *** *p* < 0.001, Student’s *t*-test.

**Table 1 ijms-24-03668-t001:** Baseline demographic characteristics of AF patients.

	ndSR	AF
Number of patients	11	9
Weight (mean ± SD) (Kg)	79.75 ± 8.62	72.01 ± 12.01
Height (mean ± SD) (cm)	168.8 ± 8.7	162.92 ± 8.06
Age (mean ± SD)	64.7 ± 9.9	70.9 ± 10.5
Sex (male/female)	9/2	6/3
Body Surface (mean ± SD) (m^2^)	1.90 ± 0.12	9
LA diameter (mean ± SD) (mm)	41.1 ± 5.9	72.01 ± 12.01

Abbreviations: LA, left atrium. LV, left ventricle.

## Data Availability

The data presented in this study are available on request from the corresponding author. The data are not publicly available due to ethical issues.

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
