# Peer review of "Increased Density of Endogenous Adenosine A_2A_ Receptors in Atrial Fibrillation: From Cellular and Porcine Models to Human Patients"

_ijms, 2023, doi:10.3390/ijms24043668_

Round 1

Reviewer 1 Report

This is an interesting manuscript aimed at defining wheter adenosine A2A receptor upregulation is a signature of atrial fibrillation. To address this question, authors developed 2 models, immortalized HL-1 myocytes (in vitro) treated with ATX-II and pigs with atrial tachypacing (in vivo), in both models they observed characteristic markers of the presence of atrial fibrillation and demonstrated upregulation of A2A receptor. Furthermore, upregulation of the A2A receptor was confirmed in biopsies of human patients with  atrial fibrillation.

The article is well written, the techniques and analysis used are appropriate, and the conclusions are well supported by results.

I only have a few minor observations:

1.- Label of Y axis is not specific; the 3 assays can be split and a suitable Y label used for each; i.e.  for PI use “number of positive nuclei”.

2.- In line 275 dichlorodihydrofluorescein is misspelled.

3.- The definition of graphs labels should be improved.

4.- It may be interesting to define the subcellular location and the functionality of A2A receptors in the models of atrial fibrillation developed in the article, if adenosine levels are increased in patients with atrial fibrillation, the receptors may be internalized.

Author Response

This is an interesting manuscript aimed at defining wheter adenosine A2A receptor upregulation is a signature of atrial fibrillation. To address this question, authors developed 2 models, immortalized HL-1 myocytes (in vitro) treated with ATX-II and pigs with atrial tachypacing (in vivo), in both models they observed characteristic markers of the presence of atrial fibrillation and demonstrated upregulation of A2A receptor. Furthermore, upregulation of the A2A receptor was confirmed in biopsies of human patients with atrial fibrillation.

The article is well written, the techniques and analysis used are appropriate, and the conclusions are well supported by results.

I only have a few minor observations:

1.- Label of Y axis is not specific; the 3 assays can be split and a suitable Y label used for each; i.e.  for PI use “number of positive nuclei”.

Response:

We agree with the reviewer that the labels of the Y axis shown in the Figure 1 are not specific. Thus, we split the three graphs incorporating a suitable Y label for each one, namely “% of cell viability”, “% of cell damage” and “% of ROS production”.  

2.- In line 275 dichlorodihydrofluorescein is misspelled.

Response:

We apologize for the inconvenience. We amended the misspelling of dichlorodihydrofluorescein throughout the manuscript since we detected another misspelling on line 79. Thus, we thank the reviewer for detecting it.

3.- The definition of graphs labels should be improved.

Response:

The reviewer is right. The label of the graphs should be better defined. Thus, we change the labels of the Y axis on graphs shown in Figures 2, 3, 4 and 5. Exactly, the former label 'integrated density' was changed to 'A2AR density' which is more precise.

4.- It may be interesting to define the subcellular location and the functionality of A2A receptors in the models of atrial fibrillation developed in the article, if adenosine levels are increased in patients with atrial fibrillation, the receptors may be internalized.

Response:

The reviewer highlighted an important issue here. From a functional point of view, we have previously shown that adenosine receptors are located near RyR2 in human atrial myocytes (CVR 2006, ref. 4) and that increased expression of A2AR in AF is associated with a higher incidence of spontaneous calcium release that can be reduced to control levels (no AF levels) by A2AR antagonists or by adenosine deaminase (EHJ 2011, ref. 4; CVR 2022, ref. 17). Together, this suggests that A2ARs remain localised near RyR2 in AF. Indeed, we now point this out in the discussion on lines 224 to 226 (“The increased spontaneous calcium release in human atrial myocytes linked to A2AR is related to an increased co-distribution of the receptor and RyR2 [4], which can be re-duced using A2AR antagonists or adenosine deaminase [4,17]”).

This is an interesting manuscript aimed at defining wheter adenosine A2A receptor upregulation is a signature of atrial fibrillation. To address this question, authors developed 2 models, immortalized HL-1 myocytes (in vitro) treated with ATX-II and pigs with atrial tachypacing (in vivo), in both models they observed characteristic markers of the presence of atrial fibrillation and demonstrated upregulation of A2A receptor. Furthermore, upregulation of the A2A receptor was confirmed in biopsies of human patients with atrial fibrillation.

The article is well written, the techniques and analysis used are appropriate, and the conclusions are well supported by results.

I only have a few minor observations:

1.- Label of Y axis is not specific; the 3 assays can be split and a suitable Y label used for each; i.e.  for PI use “number of positive nuclei”.

Response:

We agree with the reviewer that the labels of the Y axis shown in the Figure 1 are not specific. Thus, we split the three graphs incorporating a suitable Y label for each one, namely “% of cell viability”, “% of cell damage” and “% of ROS production”.  

2.- In line 275 dichlorodihydrofluorescein is misspelled.

Response:

We apologize for the inconvenience. We amended the misspelling of dichlorodihydrofluorescein throughout the manuscript since we detected another misspelling on line 79. Thus, we thank the reviewer for detecting it.

3.- The definition of graphs labels should be improved.

Response:

The reviewer is right. The label of the graphs should be better defined. Thus, we change the labels of the Y axis on graphs shown in Figures 2, 3, 4 and 5. Exactly, the former label 'integrated density' was changed to 'A2AR density' which is more precise.

4.- It may be interesting to define the subcellular location and the functionality of A2A receptors in the models of atrial fibrillation developed in the article, if adenosine levels are increased in patients with atrial fibrillation, the receptors may be internalized.

Response:

The reviewer highlighted an important issue here. From a functional point of view, we have previously shown that adenosine receptors are located near RyR2 in human atrial myocytes (CVR 2006, ref. 4) and that increased expression of A2AR in AF is associated with a higher incidence of spontaneous calcium release that can be reduced to control levels (no AF levels) by A2AR antagonists or by adenosine deaminase (EHJ 2011, ref. 4; CVR 2022, ref. 17). Together, this suggests that A2ARs remain localised near RyR2 in AF. Indeed, we now point this out in the discussion on lines 224 to 226 (“The increased spontaneous calcium release in human atrial myocytes linked to A2AR is related to an increased co-distribution of the receptor and RyR2 [4], which can be re-duced using A2AR antagonists or adenosine deaminase [4,17]”).

This is an interesting manuscript aimed at defining wheter adenosine A2A receptor upregulation is a signature of atrial fibrillation. To address this question, authors developed 2 models, immortalized HL-1 myocytes (in vitro) treated with ATX-II and pigs with atrial tachypacing (in vivo), in both models they observed characteristic markers of the presence of atrial fibrillation and demonstrated upregulation of A2A receptor. Furthermore, upregulation of the A2A receptor was confirmed in biopsies of human patients with atrial fibrillation.

The article is well written, the techniques and analysis used are appropriate, and the conclusions are well supported by results.

I only have a few minor observations:

1.- Label of Y axis is not specific; the 3 assays can be split and a suitable Y label used for each; i.e.  for PI use “number of positive nuclei”.

Response:

We agree with the reviewer that the labels of the Y axis shown in the Figure 1 are not specific. Thus, we split the three graphs incorporating a suitable Y label for each one, namely “% of cell viability”, “% of cell damage” and “% of ROS production”.  

2.- In line 275 dichlorodihydrofluorescein is misspelled.

Response:

We apologize for the inconvenience. We amended the misspelling of dichlorodihydrofluorescein throughout the manuscript since we detected another misspelling on line 79. Thus, we thank the reviewer for detecting it.

3.- The definition of graphs labels should be improved.

Response:

The reviewer is right. The label of the graphs should be better defined. Thus, we change the labels of the Y axis on graphs shown in Figures 2, 3, 4 and 5. Exactly, the former label 'integrated density' was changed to 'A2AR density' which is more precise.

4.- It may be interesting to define the subcellular location and the functionality of A2A receptors in the models of atrial fibrillation developed in the article, if adenosine levels are increased in patients with atrial fibrillation, the receptors may be internalized.

Response:

The reviewer highlighted an important issue here. From a functional point of view, we have previously shown that adenosine receptors are located near RyR2 in human atrial myocytes (CVR 2006, ref. 4) and that increased expression of A2AR in AF is associated with a higher incidence of spontaneous calcium release that can be reduced to control levels (no AF levels) by A2AR antagonists or by adenosine deaminase (EHJ 2011, ref. 4; CVR 2022, ref. 17). Together, this suggests that A2ARs remain localised near RyR2 in AF. Indeed, we now point this out in the discussion on lines 224 to 226 (“The increased spontaneous calcium release in human atrial myocytes linked to A2AR is related to an increased co-distribution of the receptor and RyR2 [4], which can be re-duced using A2AR antagonists or adenosine deaminase [4,17]”).

Reviewer 2 Report

The manuscript by Godoy-Marín et al. report the “Increased Density of Endogenous Adenosine A2A Receptors in Atrial Fibrillation: From Cellular and Porcine Models to Human Patients”.  The authors developed two experimental models of atrial fibrillation (AF). Namely, the cardiomyocyte cell line HL-1 submitted to Anemonia toxin II 26 (ATX-II) and a large animal model of AF, the right atrium tachypaced (A-TP) pig.  By checking the A2A receptor density and Csq-2 protein level, they confirmed that these two models mimic the biomarkers of AF patients similarly.

Overall, the manuscript is endowed with sufficient novelty, well-written, and could be accepted for publishing in Int. J. Mol. Sci. after minor revision followed the suggestions provided below.

1.     How are these developed models useful compared to other existing AF models?

2.     I think it would have been a better introduction if they have described about limitations of existing AF models.

3.     The current study could be recognized as an extension of previous reports (ref. 6 and ref 13 and ref 14) which are regarding the elevated APLs in AF patients and in vitro cell model of AF. So they can find the AF-induced cell model also show an increase in A2AAR expression. However, this reviewer assumed that model should have not only alteration of disease biomarker, but also showing phenotype of AF. In this paper, they didn’t mention about arrhythmic phenotype of their models.

Author Response

The manuscript by Godoy-Marín et al. report the “Increased Density of Endogenous Adenosine A2A Receptors in Atrial Fibrillation: From Cellular and Porcine Models to Human Patients”.  The authors developed two experimental models of atrial fibrillation (AF). Namely, the cardiomyocyte cell line HL-1 submitted to Anemonia toxin II 26 (ATX-II) and a large animal model of AF, the right atrium tachypaced (A-TP) pig.  By checking the A2A receptor density and Csq-2 protein level, they confirmed that these two models mimic the biomarkers of AF patients similarly.

Overall, the manuscript is endowed with sufficient novelty, well-written, and could be accepted for publishing in Int. J. Mol. Sci. after minor revision followed the suggestions provided below.

  1. How are these developed models useful compared to other existing AF models?

Response:

We thank the reviewer for this suggestion. Cultures of the HL-1 atrial cell line are commonly for testing pharmacological compounds (ref 8). Here we show that after treatment with ATX the model reproduces hallmarks of AF and exhibit similar changes in A2AR expression, demonstrating that the model could be useful for initial testing of candidate compounds targeting A2ARs. Indeed, we now point this out in the discussion on line 248 to 250 (“In general, since the ATX model shows comparable changes in A2AR expression as in AF, it could become a promising tool to evaluate candidate compounds targeting A2AR overactivity”). Subsequent validation of compounds selected based on the HL-1 model will require further testing in larger animal models or humans. As pointed out in the discussion (lines 257-268), the advantages of the porcine model are discussed (“The pig animal model of AF has gained importance because it offers several advantages: i) Pigs have anatomy and physiology similar to humans, which makes their hearts suitable for studying human heart conditions; ii) Porcine AF models have a high degree of reproducibility, making them ideal for use in preclinical trials and other research studies; iii) Development of AF through a variety of methods, such as rapid atrial pacing, transesophageal stimulation, or surgical modification; iv) The size of the porcine heart makes it possible to perform various interventional procedures, such as catheter ablation, in a manner similar to that done in humans; and v) Pigs have a longer lifespan compared to other laboratory animals, allowing longer term studies of AF. These factors make the porcine model a valuable tool for researchers studying AF and eventually for testing candidate compounds targeting the adenosinergic system.”).

  1. I think it would have been a better introduction if they have described about limitations of existing AF models.

Response:

Wea agree with the reviewer that the introduction should contain a detailed description of the pros and cons of the different AF experimental models. Thus, following the reviewer instructions we remarkably expanded the introduction considering these issues (line 64-79) (“Experimental models of AF are commonly used to study the underlying mechanisms and evaluate potential treatments for the disease. There are several types of experimental models used in AF research, including: i) animal models based on the con-trolled induction of AF in various species, such as pigs, dogs, and mice; ii) in vitro models mostly using isolated cardiomyocytes or tissue slices from various species, including humans; and iii) computational models simulating the electrical and structural properties of the heart. Each AF model has its own advantages and limitations, and the choice of model depends on the specific research question being studied. Some of the limitations of AF models include: i) lack of complete representation, thus limiting their predictive value; ii) induction methods, which may not accurately reflect the underlying mechanisms of human AF; iii) species differences (i.e., anatomy and physiology), which limits the generalisability of results from animal models to humans; iv) limited availability of human tissue because ethical considerations and the difficulty of obtaining fresh samples; v) cost and time as the maintenance and use of experimental models can be expensive and time-consuming. Despite these limitations, experimental models of AF remain an important tool for understanding the underlying mechanisms of the disease and evaluating potential treatments).     

  1. The current study could be recognized as an extension of previous reports (ref. 6 and ref 13 and ref 14) which are regarding the elevated APLs in AF patients and in vitro cell model of AF. So they can find the AF-induced cell model also show an increase in A2AAR expression. However, this reviewer assumed that model should have not only alteration of disease biomarker, but also showing phenotype of AF. In this paper, they didn’t mention about arrhythmic phenotype of their models.

Response:

The reviewer highlighted an important question here. We have previously used HL-1 myocytes (without ATX) and shown that A2AR activation favor arrhythmic responses in paced cell cultures (Ref 8). Presumably, the ATX-induced increase in A2AR expression exacerbates this response. Interestingly, we now include this point in the discussion on lines 273-278 (“Overall, the different experimental models of AF may have different arrhythmic phenotypes, as the underlying mechanisms and conditions that trigger the onset of AF can vary between models. Indeed, this can affect the features and characteristics of the arrhythmia-like phenomena observed in each model, such as the frequency, duration, and stability of the fibrillation waves. Therefore, it is important to carefully choose the appropriate experimental model to study the specific aspects of AF that are of interest”)
